# Experimental Investigation of the Flow Field in the Vicinity of an Oscillating Wave Surge Converter

**Moisés Brito** [1,2,*] **, Rui M. L. Ferreira** [1] **, Luis Teixeira** [3] **, Maria G. Neves** [4] **and Luís Gil** [2]

1   Civil Engineering Research and Innovation for Sustainability (CERIS), Instituto Superior Técnico, Universidade de Lisboa, Av. Rovisco Pais, 1049-001 Lisboa, Portugal; ruimferreira@tecnico.ulisboa.pt

2   UNIDEMI, Department of Mechanical and Industrial Engineering, NOVA School of Science and Technology, NOVA University Lisbon, 2829-516 Caparica, Portugal; lmg@fct.unl.pt

3   Instituto de Mecánica de los Fluidos e Ingeniería Ambiental (IMFIA), Facultad de Ingeniería, Universidad de la República, 11200 Montevideo, Uruguay; luistei@fing.edu.uy

4   Harbours and Maritime Structures Division, Hydraulics and Environment Department, Laboratório Nacional de Engenharia Civil (LNEC), Av. do Brasil 101, 1700-066 Lisboa, Portugal; gneves@lnec.pt

*   Correspondence: moisesbrito@fct.unl.pt

**Abstract:** The main objective of this paper is to characterize the flow field on the front face of an oscillating wave surge converter (OWSC) under a regular wave. For this purpose, the longitudinal and vertical velocity components were measured using an Ultrasonic Velocity Profiler (UVP). In order to explain the main trends of the OWSC's dynamics, the experimental data were firstly compared with the analytical results of potential theory. A large discrepancy was observed between experimental and analytical results, caused by the nonlinear behavior of wave-OWSC interaction that determine the turbulent field and the boundary layer. The experimental velocity field shows a strong ascendant flow generated by the mass transfer over the flap (overtopping) and flow rotation generated by the beginning of the flap deceleration and acceleration. These features (overtopping and flow rotation) have an important role on the power capture of OWSC and, therefore, analytical results are not accurate to describe the complex hydrodynamics of OWSC.

**Keywords:** wave energy; experimental investigation; oscillating wave surge converter (OWSC); velocity field; wave-structure interaction

## 1. Introduction

Oscillating wave surge converter (OWSC) devices are known to be competitive in the nearshore regions with water depth ranging between 10 to 20 m [1–4]. These devices are typically composed by a buoyant flap and by a hydraulic power take-off (PTO) system and are designed to exploit the enhanced horizontal fluid particle movement of waves in the nearshore regions [1,2,5]. Hence, during their operation, the flap will have large amplitudes of motion. The flap is attached to the seabed via bearings, presenting vertical position in the equilibrium position, pitching under the action of waves. The PTO system is composed by hydraulic cylinders and a closed hydraulic circuit. OWSC presents at least three stages of energy conversion [6–8]. In the first stage, the flap is excited by the wave, transforming the wave energy into mechanical energy. In the second stage, the hydraulic PTO system transforms the mechanical energy into potential energy. Finally, in the third stage, the high-pressured fluid drives a turbomachinery which converts potential energy into electrical energy [1–4].

Oscillating wave surge converter (OWSC) devices are in its early stages of development [9]. In the last decade, major advances have been registered with the implementation of some devices in a

pre-commercial stage, some of them already connected to the electric network, such as Oyster [2] and WaveRoller [3]. Full-scale prototypes of Oyster and WaveRoller have been successfully tested with unit rated of 300 kW and 315 kW, respectively. The Oyster have been developed Aquamarine Power and Queen's University Belfast, and it is installed in Orkney, Scotland. The WaveRoller was developed by AW-Energy, and it was tested in Peniche, Portugal.

The hydrodynamics of the OWSCs have been characterized with theoretical analysis, laboratory and field work, and numerical tools. Theoretical studies are usually performed using linear wave theory; see, e.g., [10–16]. Its results lead to important expressions of the hydrodynamic characteristics. However, in many cases, these solutions are not accurate to describe the complex hydrodynamics of OWSCs. A major limitation is the impossibility to account for losses due to real fluid effects including large scale turbulence and unable to accurately model large amplitudes of motion (nonlinear behaviors). Such effects are known to be important in the power capture of OWSC [7,8]. For these reasons, physical model tests are important to establish where corrections should be applied [17].

Experimental studies on the hydrodynamics of OWSCs are relatively scarce. Folley et al. [1,18] studied the geometric parameters of a 1:40 scale OWSC model using a wave flume. Their experiments have shown that the water depth has an important effect on the hydrodynamics and consequently on the performance of OWSC. Folley et al. [18] demonstrated that both the surge wave force and power capture of OWSC increase in shallow water. Whittaker et al. [6] have observed that the highest capture efficiency mostly occurs with a surface piercing flap. Henry [19] has also investigated the hydrodynamics of the OWSC at both 1:40 and 1:20 scale laboratory models. He has shown that the incident wave periods cause a marginal increase of power capture factor and the larger diameter of the freeboard reduces the viscous losses and has a greater influence on the power capture factor. The water depth and the flap width have shown a strong influence on the magnitude of wave force, and thus on the power capture factor. However, both parameters have limited effects on the hydrodynamics which reduces the power capture factor, especially in sea states with short periods. It was also observed that as the flap width increases the power capture factor gain reduces, associated with the addition of the freeboard.

The PTO damping used has shown a significant effect on the power capture factor, with a constant damping producing between 20% and 30% less power capture factor than quadratic damping. Furthermore, it was found that applying a higher level of damping, or a damping bias, the flap pitches towards the beach, increasing the power capture factor of about 10% [20]. Lin et al. [21] presented an experimental investigation on the parameters of a 1:20 scale OWSC model. They have tested and analysed different density, moment of inertia and location of center of mass of the flap in order to evaluate the performance of OWSC. Schmitt et al. [22] presented a series of experimental tests using a 1:25 scale OWSC model. A large number of wave probes were used to record free-surface elevation at different locations in the wave flume and pressure sensors were used to measure the pressure field on the faces of the flap. More recently, Henry et al. [23,24] investigated the slamming of an OWSC in extreme sea states. The physics of the slamming process were identified, and a pressure field on the front face of the flap was presented. However, in the experimental studies mentioned, the flow field around OWSCs were not measured and the phase differences between the hydrodynamic force and flow quantities such as velocity or free-surface elevation have not been investigated [4]. Furthermore, those studies do not provide the shortcomings of the analytical solutions based on the flow field.

This paper is based on novel experimental evidence, produced under controlled conditions in a laboratory set up. It addresses the issue of characterizing the flow field in front of an OWSC, highlighting the important differences between experimental and analytical results. Although there are several analytical models of hydrodynamic characteristics of OWSCs, this paper considers the model developed by Renzi and Dias [14]. This model allows the analysis of OWSC in a wave flume (i.e., takes into account the effect of sidewalls by the diffraction potential). In this paper, the free-surface elevation, pressure in the hydraulic PTO system, rotation angle, and velocity field in the vicinity of

a 10th scale OWSC model were measured and analyzed under regular waves. The longitudinal and vertical velocity components were measured using an ultrasonic velocity profiler (UVP). As UVP measures velocities along the axis of the probe, the velocity field was obtained by repeating the same test 78 times and moving the UVP probes across the vertical direction with steps of 2 cm. The velocity field was obtained by phase-averaging the velocity signal over 50 successive waves.

The present paper is organized as follows: the experimental setup and procedures, including wave flume, OWSC model, experimental apparatus, data collection, and analysis are described in Section 2; the main results are presented and discussed in Section 3; the conclusions are summarized in Section 5.

## 2. Experimental Setup and Procedures

### 2.1. Wave Flume

The experiments were performed in a flat bed wave flume at the Instituto de Mecánica de los Fluidos e Ingeniería Ambiental (IMFIA), Universidad de la República, Uruguay. The flume is 60 m long (wave direction), 1.5 m wide, and 1.8 m deep. Waves were generated by a piston-type wavemaker which is controlled by AwaSys 6 [25]. AwaSys allows for generating waves with active wave absorption. At end of the flume, there is a passive porous mesh beach with a longitudinal slope of 0.3 m/m. A schematic sketch of the side and top views of the flume arrangement is shown in Figure 1.

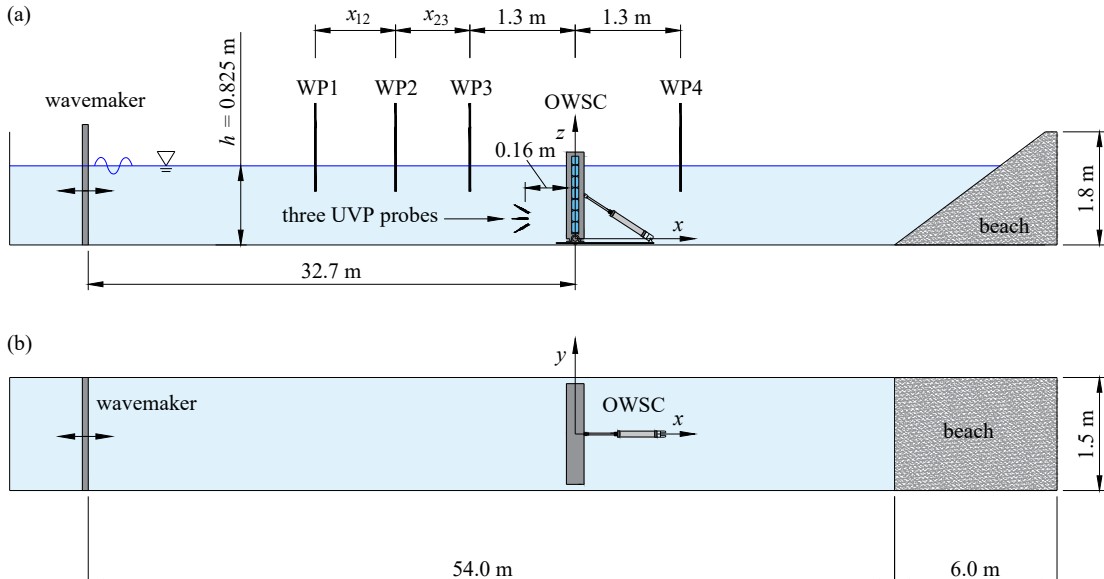

**Figure 1.** Schematic sketch of the wave flume arrangement: (**a**) side view; (**b**) plan view.

The coordinates $x$, $y$, and $z$ refer to the longitudinal (wave direction), lateral and vertical directions, respectively, for which $x = 0$ at 32.7 m from the wavemaker; $y = 0$ at center plane of the flume; and $z = 0$ at 0.1 m from the flume bed in the pivot point of the flap (see Figures 2 and 3).

### 2.2. OWSC Model

The OWSC model is composed by a buoyant flap and by a hydraulic PTO system (Figure 3a). The flap is 0.84 m height, 1.31 m wide, and 0.17 m thick. It is composed of PVC tubes (no gap between tubes was considered), a stainless steel frame, and bearings with mass $m = 72.3$ kg, moment of inertia $I = 14.74$ kg m$^2$, and center of mass located at $x = 0$, $y = 0$, and $z = 0.33$ m. The flap is attached to the foundation via bearings with internal diameter of 0.05 m, hinged on its horizontal axis at 0.10 m above the flume bed. The gap between the flume bed and the flap is 0.045 m (Figure 4). Similarly to the physical model tests of Folley et al. [1,18], Henry et al. [23,24], the OWSC was designed to represent

a quasi-2D test. Due to the physical implementation and operation, a total gap of 0.095 m between the flap and the sidewalls of the flume was considered. Even though the sidewalls will increase the available power, as noted by Count and Evans [26], a quasi-2D test can provide a greater insight into the wave-OWSC interaction.

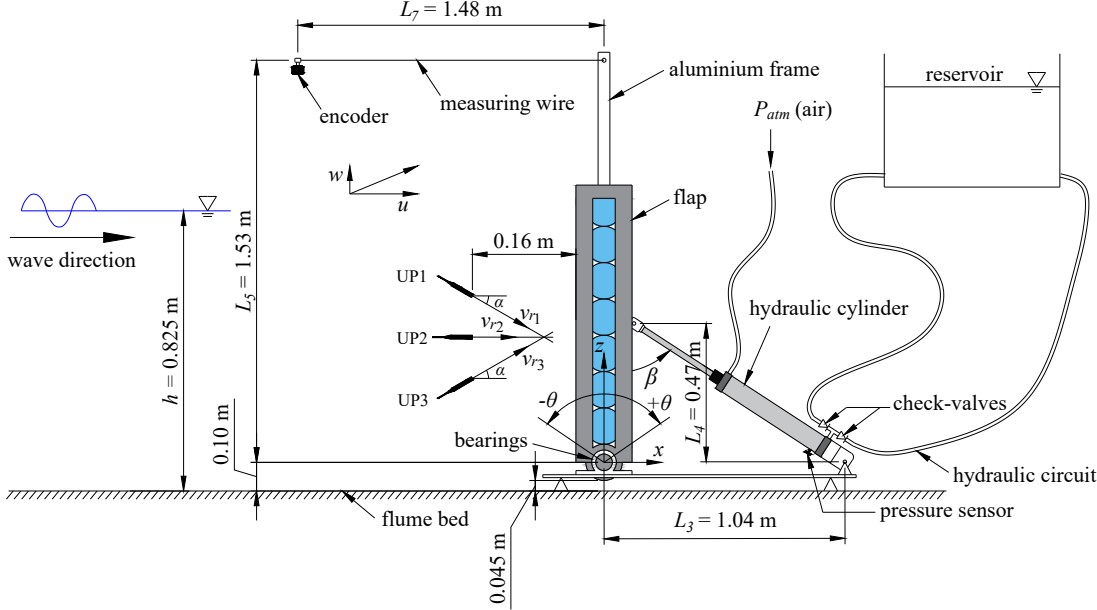

**Figure 2.** Layout of the OWSC model, illustrating: the position of the experimental setup; the configuration of the UVP probes (UP1, UP2 and UP3); and the geometry of the OWSC.

The PTO system is mainly composed of a hydraulic cylinder, valves, a hydraulic circuit, and a reservoir. In the experiments, the hydraulic cylinder was working as a single acting cylinder, with the upper cylinder chamber open to atmosphere, $P_{atm}$ (Figures 2 and 3a). The cylinder is linked to the flume bed at $z = 0$ and $x = 1.04$ m, and to the flap at $z = 0.47$ m and $x = 0.09$ m.

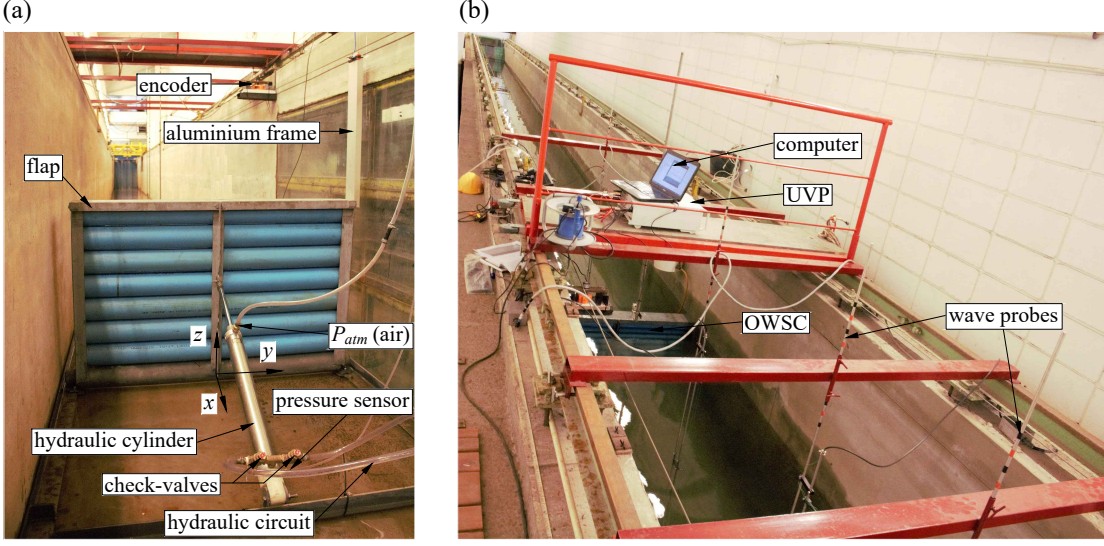

**Figure 3.** (**a**) OWSC model and (**b**) experimental apparatus.

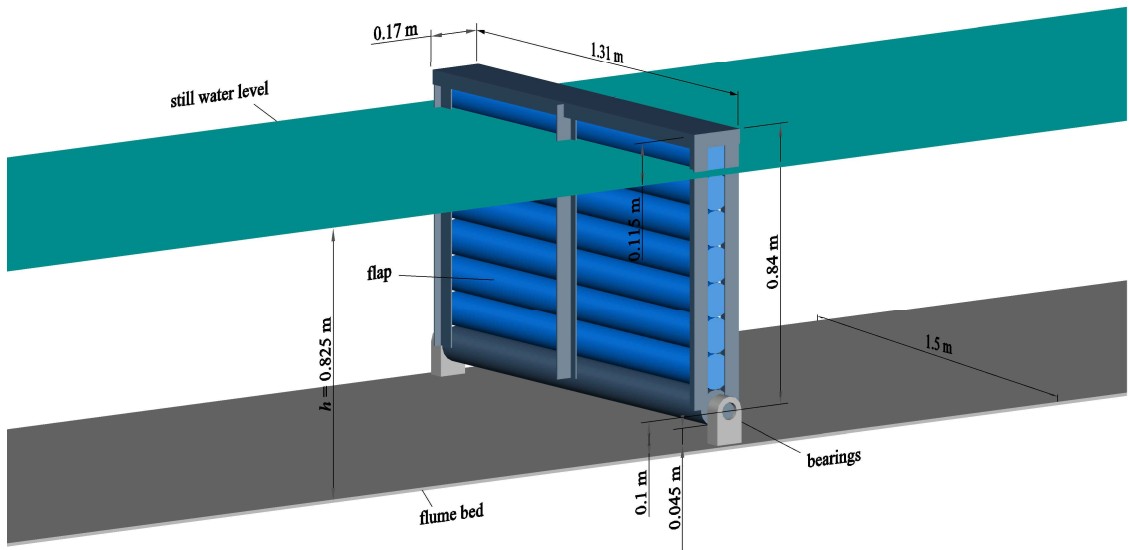

**Figure 4.** Sketch of the OWSC model in the wave flume.

### 2.3. Full Experimental Setup and Instrumentation

The layout of the OWSC model together with the experimental apparatus are shown in Figure 2.

The apparatus has been carefully placed in order to avoid any modification of the flap inertia. A linear encoder with 600 PPR (pulses per revolution) and with a 100 Hz sampling frequency was used to measure the rotation angle of the flap, $\theta$. It was placed at $x = -1.48$ m and $z = 1.53$ m and the measuring wire was linked to the flap by a vertical aluminum frame (Figure 2). The angle of the flap, $\theta$, is calculated by the following implicit equation:

$$[L_7 - \Delta L_7(t)]^2 = L_7^2 + \left[2L_5 \sin\left(\frac{\theta(t)}{2}\right)\right]^2 - 2L_7 L_5 \sin\left(\theta(t)\right) \tag{1}$$

where $t$ is the time and $\Delta L_7$ is the variation of the measuring wire length. The detailed description of the values introduced in Equation (1) can be found in Figure 2. This method was also used by Brito et al. [7].

The pressure in the cylinder chamber, $P_{int}$, was measured by a pressure sensor with an accuracy less than 0.2% RO (rated output) with a 100 Hz sampling frequency.

Four standard capacitive wave probes (WP1, WP2, WP3, and WP4 in Figure 1) produced by Akamina Technologies (Ottawa, Canada) was used to measure the wave elevation, $\eta$, in the center plane of the flume ($y = 0$). The distances between wave probes, $x_{12}$ and $x_{23}$, were chosen according to the method proposed by [27] in order to make separating the incident and reflected waves possible. In this study, five equally-spaced water submersions were used for the static calibration of the wave probes. A first-order calibration polynomial relating the output voltages to the wave elevation was obtained by a least-square fit procedure [28]. The water elevation, $\eta$, was recorded with a 25 Hz sampling frequency. In order to assess the reflection coefficient of the beach, experimental tests were also performed without OWSC. A beach mean reflection coefficient of about 8% has been found.

The flow velocity in the vicinity of the flap was measured using an Ultrasonic Velocity Profiler (UVP) produced by Met-Flow [29] in the center plane of the flume ($y = 0$). A single UVP probe only measures velocities along its own axis [30]. Therefore, to measure the longitudinal and vertical velocity components, three UVP probes (UP1, UP2 and UP3) pointing in different directions were used, as shown in Figure 2. An angle $\alpha = \pi/6$ rad between UP1, UP3, and horizontal (UP2) was considered. The UVP was operated with a sample frequency of 11.6 Hz, a Doppler frequency of 2 MHz, two cycles per pulse, and 360 channels without overlap and width of 0.74 mm. The divergence half-angle for the 2 MHz UVP transducer used is 2.2°, which gives a sampling volume with a diameter of about

2 cm close to the flap. The noise was filtered by a zero-phase digital filter with a 4th-order Butterworth low-pass filter of 2 Hz cut-off frequency [7]. The resolution of the velocity measurement is 0.031 mm/s. The velocity field was obtained by repeating the same test 78 times and moving the probes across the vertical direction with steps of 2 cm (from $z = 0.20$ to 0.72 m). The flow field is obtained by phase-averaging the velocity signal over 50 successive waves [31,32].

## 2.4. Data Collection

The experiments were carried out at 1:10 Froude's scale. Regular waves with still water depth of $h = 0.825$ m was considered. The wave conditions comprise plane progressive waves of period $T$, ranging from 2 to 4 s, and height, $H$, varying from 0.15 to 0.3 m. These conditions correspond to the highest annual frequency in the Uruguayan oceanic coast [33]. Table 1 shows the wave conditions: $x_{12}$ and $x_{23}$ considered in this study.

**Table 1.** Wave conditions used in the experimental tests.

| $T$ (s) | $H$ (m) | $x_{12}$ (m) | $x_{23}$ (m) |
| --- | --- | --- | --- |
| 2 | [0.15; 0.175; 0.2; 0.225; 0.25; 0.275; 0.3] | 0.49 | 1.23 |
| 2.25 | [0.15; 0.175; 0.2; 0.225; 0.25; 0.275; 0.3] | 0.57 | 1.42 |
| 2.5 | [0.15; 0.175; 0.2; 0.225; 0.25; 0.275; 0.3] | 0.65 | 1.62 |
| 2.75 | [0.15; 0.175; 0.2; 0.225; 0.25; 0.275; 0.3] | 0.72 | 1.81 |
| 3 | [0.15; 0.175; 0.2; 0.225; 0.25; 0.275; 0.3] | 0.8 | 2 |
| 3.25 | [0.15; 0.175; 0.2; 0.225; 0.25; 0.275; 0.3] | 0.88 | 2.19 |
| 3.5 | [0.15; 0.175; 0.2; 0.225; 0.25; 0.275; 0.3] | 0.95 | 2.38 |
| 3.75 | [0.15; 0.175; 0.2; 0.225; 0.25; 0.275; 0.3] | 1.02 | 2.56 |
| 4 | [0.15; 0.175; 0.2; 0.225; 0.25; 0.275; 0.3] | 1.1 | 2.75 |

Due to the large number of tests requested to obtained the velocity field, the flow velocity was measured for $T = 3.5$ s and $H = 0.25$ m, as this condition allows almost symmetric $\theta$ (see Section 2.5). In this condition, the reflection coefficient was about 20% [8].

## 2.5. Repeatability of Experimental Tests

As referred in Section 2.3, the velocity field was obtained by repeating the same test condition 78 times and, therefore, a high degree of repeatability of the tests is essential to ensure the accuracy of data [31,32]. Excellent repeatability for both $\eta$ and $\theta$ was observed in the experimental tests, with only small differences in the maximum amplitude of $\theta$. The standard deviations of $\eta$ amplitudes and periods were less than 0.5% and 0.2%, respectively. In the case of $\theta$ amplitudes and periods, ther were standard deviations of less than 2.1% and 0.4%, respectively. Hence, it was reasonable to assume that the tests are fully repeatable. Transient data in the time series can also cause a large distortion in the processing of statistical quantities [34]. To estimate the accurately of these quantities by means of phase-averaging, a quasi-steady condition of the tests from wave-to-wave must be ensured [32,35]. The full time series of normalized $\eta$ and $\theta$ are shown in Figure 5. Similar to Dimas and Galani [32], Ting [34], in this study, the quasi-steady condition is quantified by comparing the amplitude scale of $\eta$ and $\theta$ (dash-dotted line in Figure 5) with its maximum amplitudes in each wave cycle. The quasi-steady condition was considered when relative errors between the amplitude scale and its maximum amplitudes are less than 2%. For both $\eta$ and $\theta$, the time series reach a quasi-steady condition at $t/T \geq 35$, i.e., approximately 35 waves after the start of wave generation.

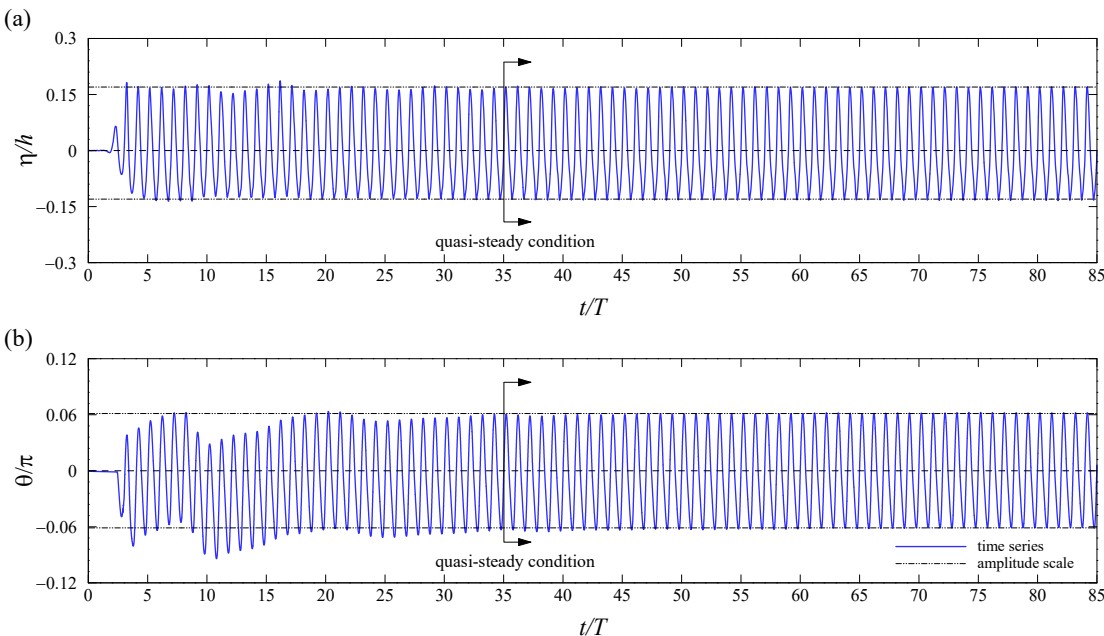

**Figure 5.** Full time series of normalized (**a**) free-surface elevation and (**b**) rotation angle of the flap. The dash-dotted line represents the amplitude scales. The vertical line at $t/T = 35$ indicates the beginning of the quasi-steady condition.

Time series of five consecutive wave periods of normalized $\eta$, $\theta$, $P_{int}$ and $\dot{\theta}$ after the quasi-steady condition, for $65 \leq t/T \leq 70$, are presented in Figure 6. The parameters presented in Figure 6 are synchronized with $\theta$ in order to analyze the phase difference of different measuring quantities.

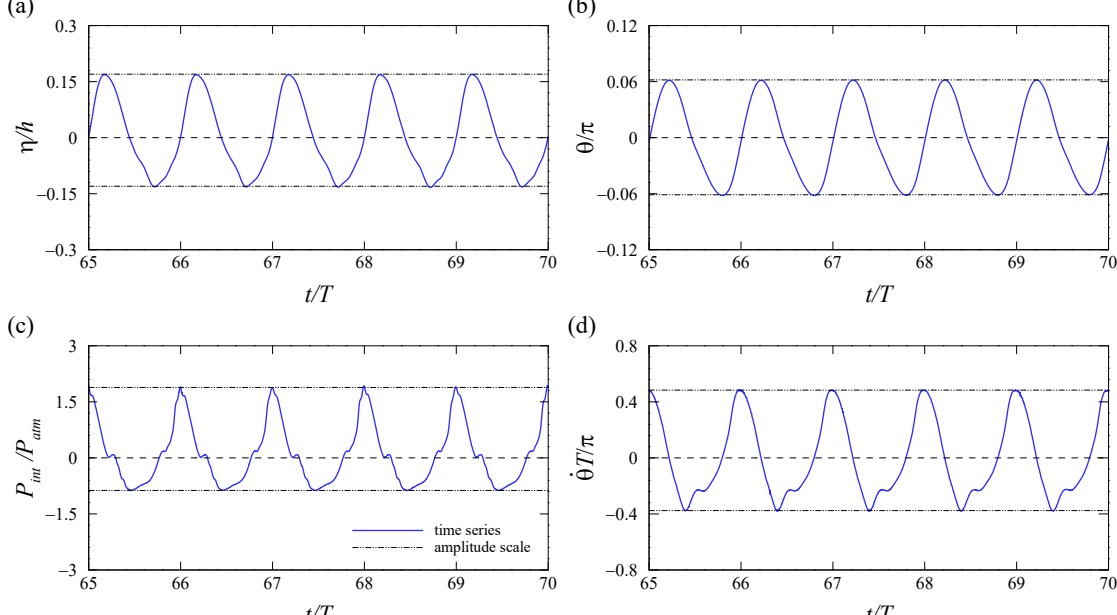

**Figure 6.** Time series for $65 < t/T < 75$ of normalized (**a**) free-surface elevation, (**b**) rotation angle of the flap, (**c**) pressure in the cylinder chamber, and (**d**) angular velocity of the flap. Dash-dotted lines represent the amplitude scale.

### 2.6. Data Analysis

In order to investigate the velocity field over a wave period, the phase-averaging was carried out [32]. The phase-averaged velocities were calculated in a quasi-steady condition over 50 wave periods (see Section 2.5). The transient data (approximately 35 waves after the start of wave generation) were removed from the time series, since these could cause large distortions in the statistical quantities [34]. The phase-averaged longitudinal, $\langle u \rangle$, and vertical, $\langle w \rangle$, velocities are given, respectively, by:

$$\langle u \rangle(x, z, t) = \frac{1}{N} \sum_{i=1}^{N} u(x, z, t + (i-1)T), \quad 0 \le t < T \tag{2}$$

and

$$\langle w \rangle(x, z, t) = \frac{1}{N} \sum_{i=1}^{N} w(x, z, t + (i-1)T), \quad 0 \le t < T \tag{3}$$

where $i$ is the oscillation cycle number and $N = 50$ is the total number of wave cycles over which averaging was performed [32,36–38].

The axial velocity, $v_r$, measured by each UVP probe, contains information of the instantaneous longitudinal, $u$, and vertical, $w$, velocity components [30] that can be expressed by:

$$v_{r1} = u_1 \cos \alpha - w_1 \sin \alpha \tag{4}$$

$$v_{r2} = u_2 \tag{5}$$

$$v_{r3} = u_3 \cos \alpha + w_3 \sin \alpha \tag{6}$$

where $v_{r1}$, $v_{r2}$, and $v_{r3}$ are the instantaneous axial velocities and the subscripts 1, 2, and 3 refer to probes UP1, UP2, and UP3, respectively (see Figure 2). $\langle u \rangle$ and $\langle w \rangle$ can be calculated by adding and subtracting Equations (4) and (6):

$$v_{r1} + v_{r3} = (u_1 + u_3) \cos \alpha + (w_3 - w_1) \sin \alpha \tag{7}$$

$$v_{r1} - v_{r3} = (u_1 - u_3) \cos \alpha - (w_1 + w_3) \sin \alpha \tag{8}$$

Averaging and assuming that the flow is statistically uniform over the vertical and longitudinal direction (i.e., $\langle u_1 \rangle = \langle u_3 \rangle = \langle u \rangle$ and $\langle w_1 \rangle = \langle w_3 \rangle = \langle w \rangle$), yields to:

$$\langle u \rangle = \frac{\langle v_{r1} \rangle + \langle v_{r3} \rangle}{2 \cos \alpha} \tag{9}$$

$$\langle w \rangle = \frac{\langle v_{r3} \rangle - \langle v_{r1} \rangle}{2 \sin \alpha} \tag{10}$$

## 3. Laboratory Evidence and Linear Wave Theory

### 3.1. Dynamic Equilibrium

The equation of the motion of the flap expresses the dynamic equilibrium of the torque about the bearings, and it is given by:

$$T_h = I\ddot{\theta} + T_g + T_{PTO} \tag{11}$$

where $I$ is the moment of inertia, $\ddot{\theta}$ is the angular acceleration of the flap, $T_h$ is the hydrodynamic torque, $T_g$ is the torque due to the gravity, and $T_{PTO}$ is the torque exerted on the flap by the PTO system, which is given by:

$$T_{PTO} = T_f + T_p \tag{12}$$

where $T_f$ is the torque due to friction force and $T_p$ is the torque due to pressure forces of the hydraulic cylinder, given by $T_f = F_f \sin \beta$ and $T_p = F_p \sin \beta$, where $\beta$ is the angle between the piston rod and the flap (see Figure 2 for details), and $F_f$ is the friction force of the hydraulic cylinder, calculated according to Appendix B, and $F_p$ is the pressure force of the hydraulic cylinder that can be given by the following model: [7]:

$$F_p = A \begin{cases} K_p \dot{\theta}^2 + I_p \ddot{\theta} & \text{for } \sigma \geq \sigma_c \text{ (no cavitation)} \\ P_v + I_p \ddot{\theta} & \text{for } \sigma < \sigma_c \text{ (cavitation)} \end{cases} \tag{13}$$

where $A$ is the cross-section area of the cylinder chamber, and $K_p$ and $I_p$ are respectively the coefficient of pressure loss and the inertia of the fluid [7], $P_v$ is the vapor pressure of water, $\sigma_c$ is the critical cavitation coefficient, and $\sigma$ is the Thoma coefficient, defined as:

$$\sigma = \frac{P_{atm} + P_{int} - P_v}{P_{atm}} \tag{14}$$

In this paper, it was assumed that $\sigma_c = 0$ as in Brito et al. [7]. The comparison between the measured and the fitted $T_p$ for $T = 3.5$ s and $H = 0.25$ m, and for $T = 2.5$ s and $H = 0.2$ m is shown in Figure 7. The fitted $T_p$ shows a good agreement with the least squares goodness-of-fit $R^2 = 0.95$. This model predicts the dynamic behaviors of $T_p$ with a satisfactory accuracy, and therefore allows for estimating $T_h$ from Equation (11).

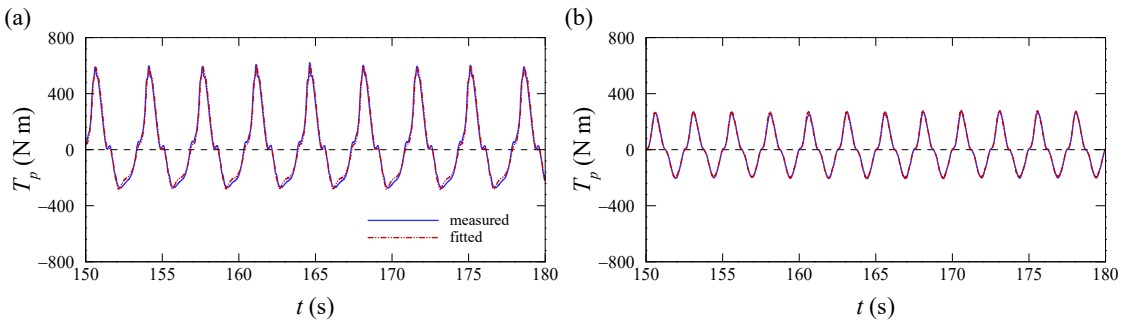

**Figure 7.** Comparison between measured and fitted torque due to the pressure force of the hydraulic cylinder for (**a**) $T = 3.5$ s and $H = 0.25$ m, (**b**) $T = 2.5$ s and $H = 0.2$ m.

### 3.2. Hydrodynamic Torque: Experimental and Analytical Results

The variation of the amplitude of hydrodynamic torque, $T_h$, with $T$ for $H = 0.15, 0.2$ and $0.25$ m is shown in Figure 8. This value was obtained by the phase-averaging of maximum and minimum hydrodynamic torque in each period. The experimental results are plotted against the analytical results of Renzi and Dias [14]. Although the order of magnitude of $T_h$ from the measurements is similar to that of the analytical solution and a decreasing trend can be envisaged, there are significant discrepancies between experimental and the analytical results. The most significant differences are found for higher wave periods, especially in the case of higher values of $H$. This behavior is caused by the nonlinearities of the interactions of the wave with the flume bed and by the interactions between the wave and the OWSC. In both cases, these nonlinear interactions become more relevant as $H$ increases.

Figure 9 shows the variation of the amplitude of $\theta$ and the power capture factor, $C_f$, with $T$ for the same conditions presented in Figure 8. The experimental $C_f$ is defined as $C_f = \overline{W}_{out}/W_I$, where $\overline{W}_{out}$ is the mean power capture and $W_I$ is the period-average incident energy flux times the width of the flap, calculated according to linear wave theory (see Appendix A for details). The instantaneous $W_{out}$ is calculated by $W_{out} = P_{int}Q$, where $Q$ is the flow rate in the hydraulic PTO system. A large discrepancy can be observed between experimental and analytical results, featuring a much smaller variation of $\theta$ and $C_f$ in the experiments. This large discrepancy can be explained by the experimental asymmetry of the OWSC motion, caused by a wave-flume bed (due to the shallow water waves) and by wave-OWSC interactions.

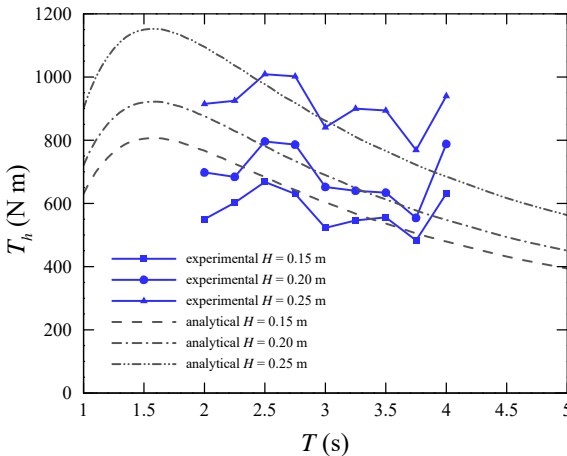

**Figure 8.** Comparison between experimental and analytical variations of the amplitude of hydrodynamic torque with a wave period.

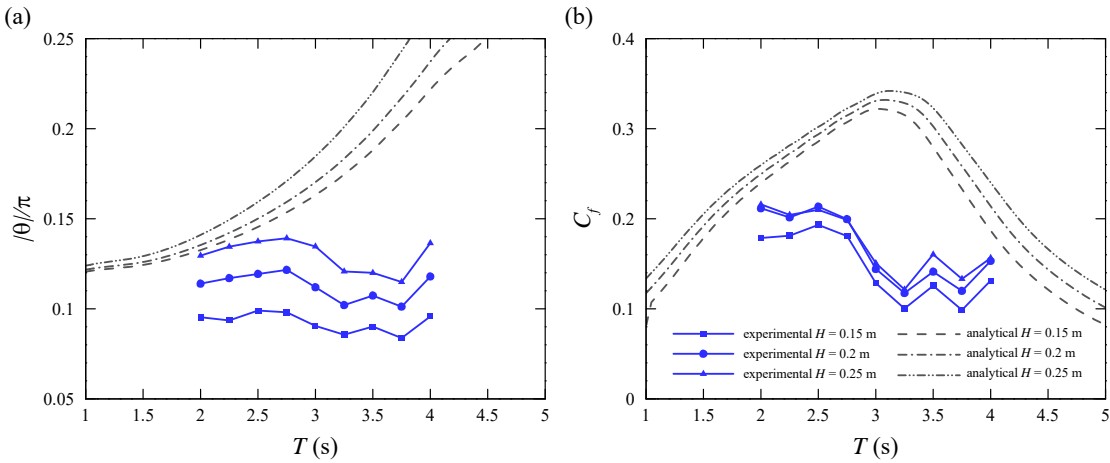

**Figure 9.** Comparison between experimental and analytical variation of (**a**) amplitude of rotation angle of the flap and (**b**) power capture factor with the wave period.

### 3.3. Dynamics of OWSC: Comparison between Experimental and Analytical Results

The normalized phase-averaged of free-surface elevations, $\langle \eta \rangle$, rotation angle of the flap, $\langle \theta \rangle$, power capture, $\langle W_{out} \rangle$, and angular velocity of the flap, $\langle \dot{\theta} \rangle$, are shown in Figure 10. In terms of wave cycle, at $t/T = 0$, the flap is in the vertical position with $\langle \theta \rangle = 0$. As the wave crest approaches, the flap moves toward the beach, with $\langle \theta \rangle / \pi > 0$, reaching its maximum $\langle \theta \rangle / \pi \approx 0.06$, being fully submerged as the wave crest passes over it at $t/T \approx 0.2$. Once the flap is fully submerged, the angular velocity decreases rapidly and the flap stops. As the wave trough approaches, the wave pressure is reduced and the water level begins to drop and the flap rises up. As the flap reaches the vertical position, $\langle \theta \rangle / \pi = 0$, and, as the wave is in the trough phase, it is moving towards the wavemaker with $\langle \theta \rangle / \pi < 0$. The water level is low providing a little resistance to the flap motion. The association between low water level and strong nonlinear behavior of the PTO system causes the steep gradient of $\langle \dot{\theta} \rangle$ at $0.4 < t/T < 0.6$ (Figure 10d). At $t/T \approx 0.75$, the flap reaches its minimum $\langle \theta \rangle / \pi \approx -0.06$. It should be noted that this minimum $\langle \theta \rangle$ occurs after the minimum of $\langle \eta \rangle$ due to the inertia of the system. As the water level increases, the wave pressure increases and the flap starts to move towards the beach and the cycle is repeated.

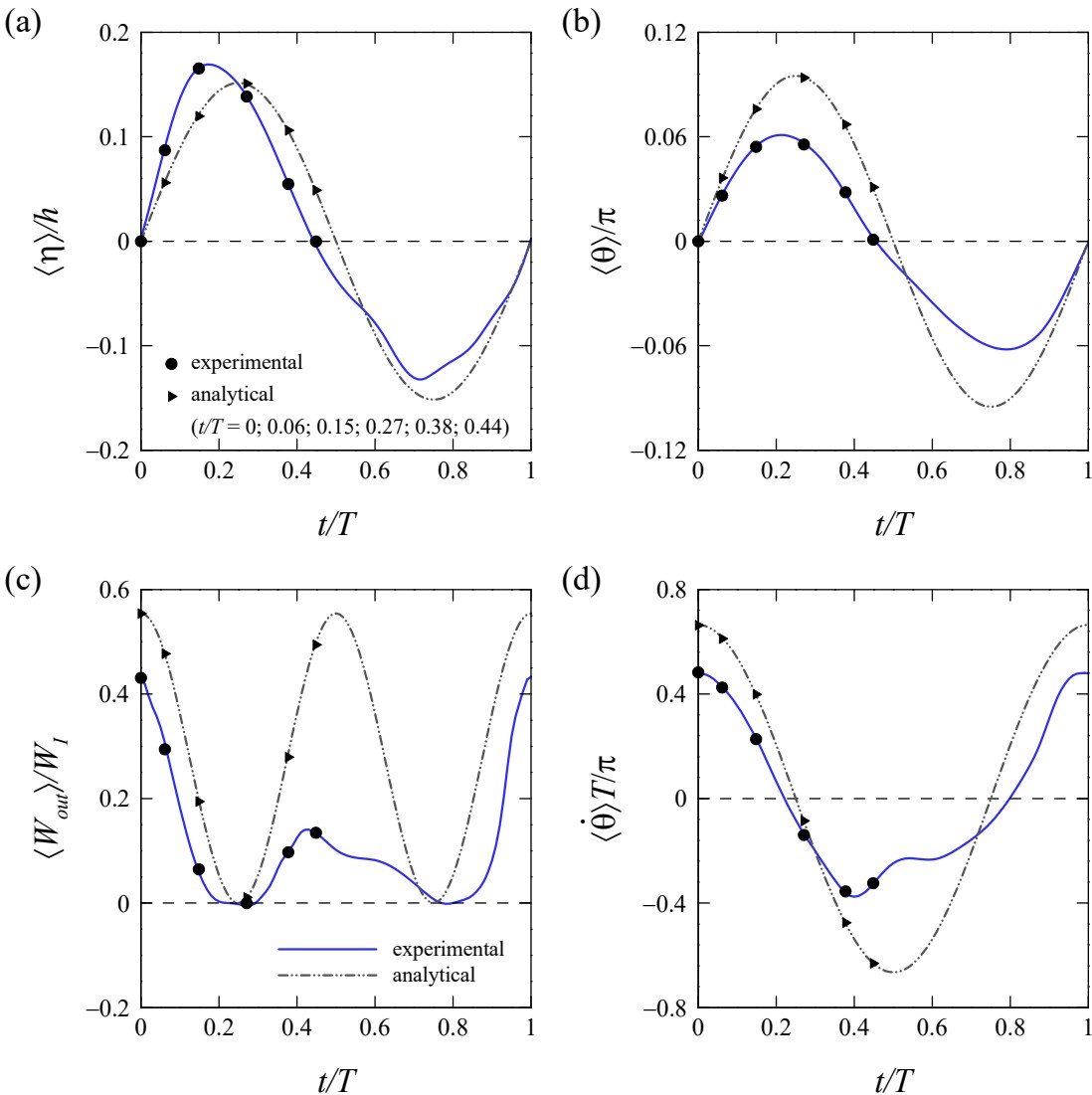

**Figure 10.** Comparison between experimental and analytical phase-averaged of normalized (**a**) free-surface elevation, (**b**) rotation, (**c**) power capture, and (**d**) angular velocity of the flap.

As depicted in Figure 10, the experimental data show a strong nonlinear behavior. In general, due to nonlinearity of the wave-OWSC interaction, a large discrepancy can be observed between analytical and experimental results. The experimental wave crest occurs at $t/T \approx 0.2$ and the analytical at $t/T = 0.25$. This experimental asymmetry is caused by the interactions of wave with the flume bed and by the interactions between waves and the OWSC. Such interactions have a more pronounced influence on $\langle\theta\rangle$, $\langle W_{out}\rangle$ and $\langle\dot\theta\rangle$. The maximum $\langle\theta\rangle$ is 37% smaller than the analytical one (given by Equation (A1)). As for $\langle\eta\rangle$, the maximum experimental absolute value of $\langle\theta\rangle$ does not match with the time of maximum analytical $\langle\theta\rangle$. The experimental $\langle W_{out}\rangle$ shows a large discrepancy from the analytical (Figure 10c), with $\langle\overline{W}_{out}\rangle/W_I = 0.12$ and 0.33, respectively. Such discrepancy is generated by the nonlinear characteristics of the PTO system and flow field. The interval when $\langle\theta\rangle/\pi < 0$ is characterized by a steep gradient of $\langle\dot\theta\rangle$ due to the PTO system and due to the rapid variations and a complete change in the nature of the flow, as seen in next section and in Figure 11.

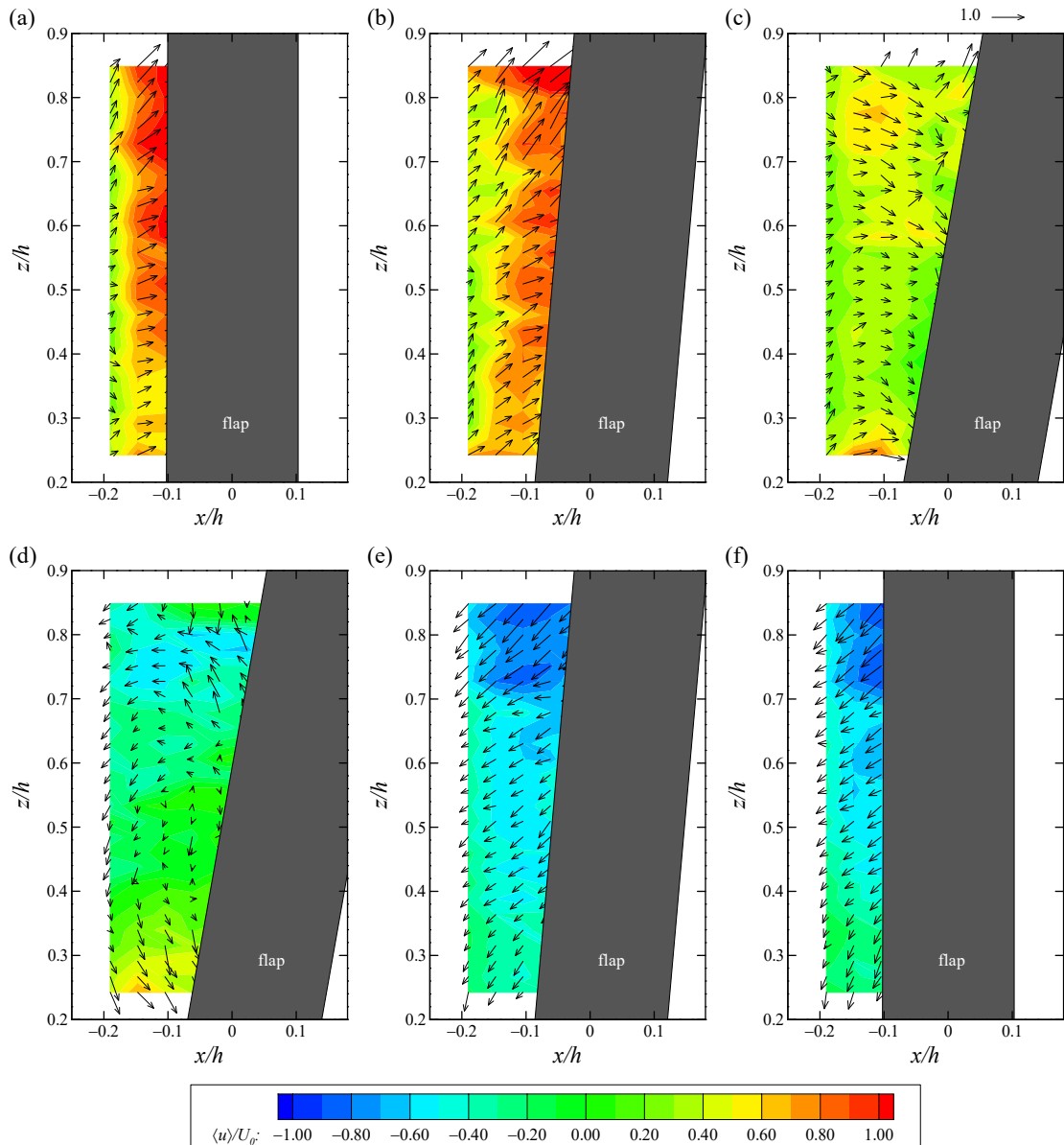

**Figure 11.** Experimental phase-averaged velocity vector field and contour of longitudinal velocity normalized by $U_0$ at (**a**) $t/T = 0$, (**b**) $t/T = 0.06$, (**c**) $t/T = 0.15$, (**d**) $t/T = 0.27$, (**e**) $t/T = 0.38$, and (**f**) $t/T = 0.44$.

## 4. Mean and Turbulent Flow Field in Front of the OWSC

The differences between experimental and analytical results are hereby analyzed. To allow for a deeper understanding of the insufficiency of the linear wave theory, mean flow field and turbulent quantities are presented and discussed. The experimental phase-averaged velocity vector field, $(\langle u \rangle, \langle w \rangle)$, and contour of $\langle u \rangle$ normalized by the deep-water maximum particle velocity, $U_0 = \pi H/T$, in the points presented in Figure 10b ($t/T = 0$, 0.06, 0.15, 0.27, 0.38, and 0.44) are shown in Figure 11. The considered $t/T$ refers to the half wave cycle with $0 \le t/T \le 0.44$. It should be noted that data for $t/T > 0.44$ are not shown as the flap covers most of the experimental mesh. In terms of $(\langle u \rangle, \langle w \rangle)$ field, a difference in the orientation of velocity near the flume bed and free-surface is observed at $t/T = 0$. For example, for $z/h < 0.5$, the vectors are nearly horizontal, with $\langle w \rangle < \langle u \rangle / 2$, and for $z/h > 0.7$ the vectors present strong upward $\langle w \rangle$, with $\langle w \rangle \approx \langle u \rangle$. This increase of $\langle w \rangle$ with an increasing of $z$ is an expression of the ascendant flow which is caused by the blockage effect of the flap when the wave crest approaches [26]. Hence, from $t/T = 0$ to 0.06, the vertical acceleration is larger than the longitudinal

one. In fact, the flow presents longitudinal deceleration at $z/h > 0.6$ (see profiles in Figure 12a,b). At $t/T = 0.06$, the vectors are markedly oriented towards the free-surface, indicating that practically the entire mass of water passes over the flap. The interval $0.06 \leq t/T \leq 0.15$ is characterized by rapid variations and a complete change in the nature of the flow. Such large deceleration is not surprising since $\langle \dot{\theta} \rangle$ evolves very rapidly (Figure 10d) and the flow decelerates to compensate the pressure field. Another key feature at $t/T = 0.15$ is the flow rotation generated by the beginning of the deceleration. From $t/T = 0.15$ to 0.27, a largest flow deceleration can be observed, caused by the arrival of the wave crest at $t/T \approx 0.2$ and the change on the orientation of $T_h$. This can be seen in the large deceleration visible in Figure 11c and d as well as in the velocity profiles (Figure 12), which are both irreconcilable with potential flow description, with the main difference due to flow rotation. For $t/T \geq 0.27$, the vectors were always oriented toward the wavemaker, indicating that the surrounding water is moving with $\langle u \rangle / U_0 < 0$. However, near the flume bed ($z/h < 0.4$), $\langle u \rangle / U_0 > 0$, confirming the flow rotation (Figure 12d). The fully descendant flow with $\langle w \rangle \approx \langle u \rangle$ is observed at both $t/T = 0.38$ and 0.44, with small deceleration from $t/T = 0.38$ to 0.44. At the instants, similar orientation of $(\langle u \rangle, \langle w \rangle)$ can be explained by the lower variation of $\langle \dot{\theta} \rangle$ (Figure 10d).

In terms of $\langle u \rangle$ contours, at $t/T = 0$ the maximum magnitude of $\langle u \rangle$ is located near the free-surface. The position of this maximum $\langle u \rangle$ is generated by the wave crest and the maximum $\langle \dot{\theta} \rangle$. From $t/T = 0$ to 0.06, the flow presents longitudinal acceleration near the flume bed, for $z/h < 0.4$, and deceleration near the free-surface at $z/h > 0.6$. At $t/T = 0.06$, $\langle u \rangle$ shows a longitudinal gradient, due to the differences in the turbulent field, with increases of turbulent kinetic energy near the free-surface and close to the flap. In this accelerating phase, turbulence is generated by the shear instability at a slight distance from the flap but is suppressed and cannot develop due to the viscous boundary layer. Naturally, the envisaged presence of the thin viscous sub-layer (not measured) in the flap should prevent too extreme accelerations. As mentioned above, the flow at $0.15 \leq t/T \leq 0.27$ is characterized by rapid variations and a complete change in its nature. At this phase, the beginning of flow deceleration, turbulence production is much enhanced and is maintained by the braking of flap motion in the maximum $\theta$. Consequently, the spatial distribution of $\langle u \rangle$ is complex and is slightly ambiguous, with the gradients, related to the deceleration of the flow. At this interval the dominant feature is a nearly constant $\langle u \rangle$ at $z/h < 0.4$ and the association between $\langle u \rangle / U_0 < 0$ and $\langle u \rangle / U_0 > 0$, and, therefore, $\langle u \rangle$ provide a clear pattern of flow rotation. In Figure 11e,f, $\langle u \rangle$ shows lower longitudinal and vertical acceleration and deceleration than those observed at $t/T \leq 0.27$.

The $\langle u \rangle$ and $\langle w \rangle$ profiles normalized by $U_0$ at section $x/h = -0.11$ and at $t/T = 0, 0.06, 0.15, 0.27$, 0.38 and 0.44 are presented in Figure 12. As expected, $\langle u \rangle$ and $\langle w \rangle$ profiles present a strong phase variation. At $t/T = 0$, the magnitude of $\langle w \rangle$ is clearly smaller than the magnitude of $\langle u \rangle$, and as the flap moves towards the beach, the variation of $\langle u \rangle$ is much lower than $\langle w \rangle$. This higher variation of $\langle w \rangle$ indicates that the turbulence and the ascendant flow generated by the mass transfer over and below the flap have small influence on $\langle u \rangle$. However, from $t/T = 0.15$ to 0.27, the variation of $\langle u \rangle$ is considerably higher than $\langle w \rangle$, indicating that the deceleration and the acceleration have a large influence on $\langle u \rangle$. Small longitudinal acceleration and deceleration can be observed in Figure 12f, where $\langle u \rangle$ profile at $t/T = 0.38$ is similar to the one at $t/T = 0.44$. However, the $\langle w \rangle$ profile presents some differences near the flume bed ($z/h \approx 0.3$) due to the return flow below the flap occurring to maintain mass conservation in the flume. At $t/T \geq 0.38$, similar shapes of $\langle w \rangle$ and $\langle u \rangle$ were observed.

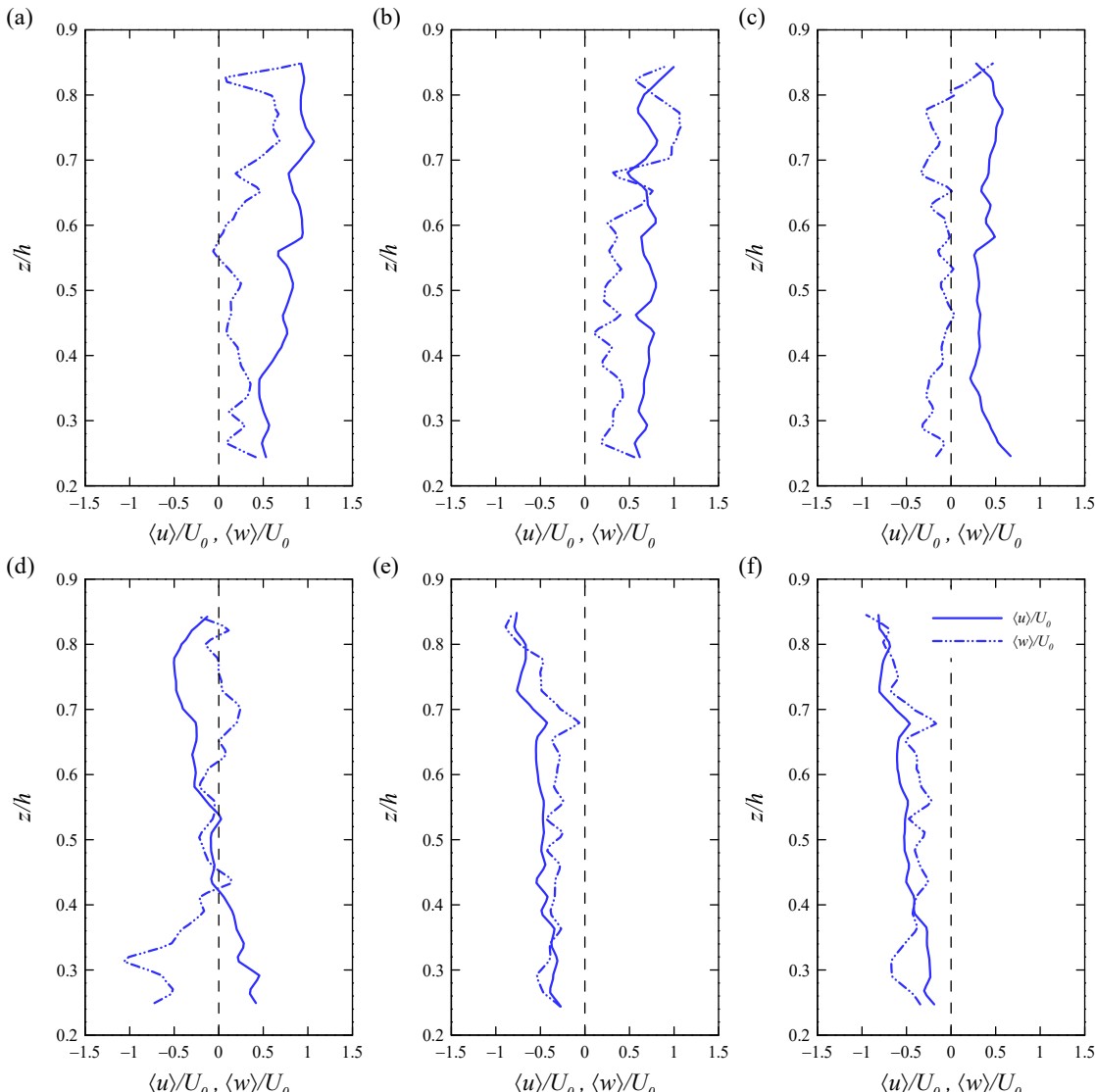

**Figure 12.** Longitudinal and vertical phase-averaged velocity profiles normalized by $U_0$ at $x/h = -0.11$ and (**a**) $t/T = 0$, (**b**) $t/T = 0.06$, (**c**) $t/T = 0.15$, (**d**) $t/T = 0.27$, (**e**) $t/T = 0.38$, and (**f**) $t/T = 0.44$.

## 5. Conclusions

This paper is based on novel experimental evidence, produced under controlled conditions in a laboratory setup. It addresses the issue of characterizing the flow field in front of an oscillating wave surge converter (OWSC), highlighting the important differences between experimental and analytical results, caused by the nonlinear behaviors of wave-OWSC interaction that govern the flow field and the boundary layer instead of the inviscid and irrotational flow.

The velocity field shows a strong ascendant flow, generated by a mass transfer over the flap, due to the approach of the wave crest and the blockage effect of the flap. The velocity vectors are markedly oriented towards the free-surface, indicating that practically the entire mass of water passes over the flap and hence the vertical acceleration is mostly larger than longitudinal one (in fact, the flow presenting longitudinal deceleration). The interval when the flap is in the maximum rotation angle is characterized by rapid variations and a complete change in the nature of the flow. Such large deceleration is not surprising since angular velocity of the flap evolves very rapidly and the flow decelerates to compensate the pressure field.

Another key feature of velocity field is the flow rotation generated by the beginning of the flap deceleration. The largest deceleration of the flow can be observed, caused by the arrival of the wave

crest and the change of the orientation of hydrodynamic torque. The flow rotation observed in the velocity field generated by wave-OWSC interaction and mass transfer have an important role on the power capture of OWSC and, therefore, analytical results are not accurate to describe the complex hydrodynamics of OWSC observed in the experimental data.

**Author Contributions:** M.B., R.M.L.F., L.T. and M.G.N. conceived and designed the experiments; M.B. performed the experiments; M.B. and R.M.L.F. analyzed the data; R.M.L.F., L.T. and M.G.N. contributed analysis tools; M.B., R.M.L.F., L.T., M.G.N. and L.G. wrote the paper. All authors have read and agreed to the published version of the manuscript.

**Funding:** This work was funded nationally through the Portuguese Foundation for Science and Technology (FCT) project PTDC/CTAOHR/30561/2017.

**Acknowledgments:** The first author acknowledges IMFIA for allowing the realization of the physical model experiments and the support of FCT through Grant No. PD/BD/705970/2014.

**Conflicts of Interest:** The authors declare no conflict of interest.

## Abbreviations

The following abbreviations are used in this manuscript:

| | |
|---|---|
| OWSC | oscillating wave surge converter |
| PTO | power take-off |
| UVP | ultrasonic velocity profiler |
| WP | wave probe |
| UP | ultrasonic velocity profiler probe |
| RO | rated output |
| PPR | pulses per revolution |

## Appendix A. Analytical Hydrodynamic Parameters

The flap rotation angle is given by $\theta(t) = \Re\{\Theta e^{-i\omega t}\}$, where $\Re$ is the real part and $\Theta$ is the complex amplitude of $\theta$ given by:

$$\Theta = \frac{T_h}{-\omega^2(I + \mu) + C - i\omega(\nu + \nu_{PTO})} \tag{A1}$$

where $\omega = 2\pi/T$ is the angular frequency of the flap, $\mu$ is the added inertia of the flap, $\nu$ is the radiation damping, $\nu_{PTO}$ is the coefficient of PTO damping, and $C = 605$ N m is the torque due to net buoyancy of the flap [14,16]. The parameters $\mu$, $\nu$, $T_h$, and $\nu_{PTO}$ were calculated following [14]. The angular velocity of the flap is defined as $\dot{\theta} = i\omega\theta$. In this study, the numerical solution of Equation (A1) was performed using a MATLAB code and using the numerical coefficients presented in [14].

The analytical power capture is calculated as $\langle W_{out}\rangle = \nu_{PTO}\langle\dot{\theta}\rangle^2$ and the period-average incident energy flux times the width of the flap, $W_I$, is calculated by:

$$W_I = \frac{1}{8}\rho g H^2 \frac{\omega}{2k}\left(1 + \frac{2kh}{\sinh 2kh}\right) L \tag{A2}$$

where $L = 1.31$ m is the width of the flap, $\rho = 1000$ kg m$^{-3}$ is the water density, $g = 9.81$ m s$^{-2}$ is the gravitational acceleration, and $k$ is the wave number given by the dispersion relationship $\omega^2 = gk\tanh(kh)$.

## Appendix B. Friction Force Model

The friction force is given by the modified LuGre model [7,39,40]:

$$F_f = \sigma_0 z + \sigma_1 \dot{z} + \sigma_2 \dot{x} \tag{A3}$$

and

$$\dot{z} = \dot{x} - \frac{\dot{x}}{g_s(\dot{x}, h_l)}z \tag{A4}$$

where $z$ is the mean deflection of the elastic bristles, $h_l$ is the dimensionless fluid film thickness parameter, $\dot{x}$ is the linear relative velocity between the piston rod and the cylinder, $\sigma_0$ and $\sigma_1$ are the dynamics coefficients, standing for stiffness and micro-viscous friction coefficient of the bristles, respectively. The static coefficient $\sigma_2$ stands for the viscous friction coefficient. This model takes the Coulomb friction force, $F_c$, and Stribeck effect in consideration through the Stribeck function, $g_s$, given by:

$$g_s(\dot{x}, h_l) = \frac{1}{\sigma_0}[(1 - h_l)F_c + (Fs - Fc)]e^{-(|\dot{x}|/|\dot{x}_s|)^n} \tag{A5}$$

where $h_l$ can be expressed as

$$\dot{h}_l = \frac{1}{\tau_h}[h_{ss} - h_l] \tag{A6}$$

with

$$\tau_h = \begin{cases} \tau_{hp} & \{\dot{x} \neq 0, h_l \leq h_{ss}\} \\ \tau_{hn} & \{\dot{x} \neq 0, h_l > h_{ss}\} \\ \tau_{h0} & \{\dot{x} = 0\} \end{cases} \tag{A7}$$

$$h_{ss} = \begin{cases} K_f|\dot{x}|^{2/3} & \{|\dot{x}| \leq |\dot{x}_b|\} \\ K_f|\dot{x}_b|^{2/3} & \{|\dot{x}| > |\dot{x}_b|\} \end{cases} \tag{A8}$$

$$K_f = \left(1 - \frac{F_c}{F_s}\right)|\dot{x}_b|^{2/3} \tag{A9}$$

where $h_{ss}$ is the dimensionless steady-state of $h_l$, $K_f$ is the proportional constant, $\dot{x}_b$ is the velocity at which the steady-state of $F_f$ becomes minimum, $\dot{x}_s$ is the Stribeck velocity, $n$ is the exponent of $g_s$ curve, $F_s$ is the maximum static friction force and $\tau_{hp}$, $\tau_{hn}$, and $\tau_{h0}$ are the time constants for acceleration, deceleration, and dwell periods, respectively. In Equation (A7), $h_l < h_{ss}$ corresponds to the acceleration period and $h_l > h_{ss}$ to the deceleration period. It is assumed by Equation (A8) that the $h_l$ is increased with $\dot{x}$ only in the negative resistance regime, $|\dot{x}| \leq |\dot{x}_b|$, and is kept at a maximum value outside this regime. All parameters and constants intruded here can be found in [7].

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
