# Peer review of "Experimental Investigation of the Flow Field in the Vicinity of an Oscillating Wave Surge Converter"

_jmse, doi:10.3390/jmse8120976_

Round 1

Reviewer 1 Report

This paper aims to investigate the hydrodynamic interactions of an oscillating wave surge converters (OSWC), in regular wave conditions.

This paper is well-organised and well-written. The phenomena studied in the paper represent a class of important wave energy problems. Although the authors did not bring modelling tools (e.g. CFD based modelling method) to reduce the model mismatch, the paper still made a contribution to the wave energy research.

Hydrodynamic modelling via linear wave theory does have limitations, e.g. the loss of fidelity when the OSWC is oscillating in large motion. This discrepancy between the (linear wave) theoretical results and the experimental results is clearly demonstrated. The experimental setup is clear.

The result matches our expectations. In large oscillating, the nonlinear effects, which are not modelled using the linear wave potential approach, play a vital role in reducing the motion, and thus, reduce the energy output. In future work, the authors are encouraged to compare the experimental results with the CFD-modelling method.

Some minor issues need to be fixed, e.g. Inconsistent citation format.

Author Response

Our answers to Reviewer 1 are in the attached file.

Reviewer 2 Report

The material is proposing a sophisticated experimental procedure. You should emphasis more the importance of experiment. The time series of measured  and <v> at the any depth should be indicated.

  • Introduction: OWSC has been studied several years. It is already installed in any coast? Please make a short introduction of development stage of OWSC.
  • Line 123: The explanation about UVP system is too short. Please indicate us the attachment, noise handling, relative angle, etc.
  • Figure 6,7 ; The error between experimental and analytical variation is too large. Please express the simple reason of that. The variation of experiment becomes near to uniform. Please indicate the reason why the variation is small as the time changes ,
  • Figure 10; The experimental curve is made by the connection line of the marked symbols? If so, how the curve is plotted in the range of t/T > 0.5?
  • Figure 11; Indication for the intensity of velocity vector is necessary.
  • In the conclusions, how is the intensity of flow on the OWSC surface useful to designing the system efficiency?

Author Response

Our replies to Reviewer 2 are in the attached file.

Reviewer 3 Report

Journal of Marine Science and Engineering

Experimental investigation of the flow field in the vicinity of an oscillating wave surge converter

by Moisés Brito, Rui M. L. Ferreira, Luis Teixeira, Maria G. Neves and Luís Gil

Reviewer’s report

The paper presents the experimental results aimed to investigate the flow on the front of an oscillating wave surge converter under regular waves.

The argument is of some interest for the scientific community and can be published after a major revision.

Specific comments:

  1. Please improve the description of the UVP system: size, precision, accuracy, control volume size, number of points in each profile, etc.
  2. Please provide some extra information about the experiments (i.e. the reflection coefficient).
  3. Please better justify the statements on the influence of turbulence also in relation to the phase average you have made.

Minor point

  1. I assume you use Latex to write the paper. Please use the “citet” command when cite a paper within a sentence (i.e L. 35, 38, 39, 51, 53, 56, 66, 95, 97, 111, 116, 123, 156, 183)
  2. 46 – Please add “flap” before width
  3. 135-135 – Please indicate the model scale for which the indicated parameters represent the conditions corresponding to the highest annual frequency in the Uruguayan oceanic coast.
  4. 144 – eq. 11 - Please describe the meaning of “I”

Author Response

Our replies to Reviewer 3 are in the attached file.

Round 2

Reviewer 2 Report

The pointed parts are well revised. The revised version is very understandable.

The only one revision opinion is follows;

Line 64: Please insert the explanation on Sec.4.

Reviewer 3 Report

Journal of Marine Science and Engineering

Experimental investigation of the flow field in the vicinity of an oscillating wave surge converter

by Moisés Brito, Rui M. L. Ferreira, Luis Teixeira, Maria G. Neves and Luís Gil

Reviewer’s report

The authors replied to all my previous issues. Therefore, the paper can be published in present form.